# High Expression of KLF10 Is Associated with Favorable Survival in Patients with Oral Squamous Cell Carcinoma

**DOI:** 10.3390/medicina57010017

**Published:** 2020-12-28

**Authors:** Chung-Min Yeh, Yi-Ju Lee, Po-Yun Ko, Yueh-Min Lin, Wen-Wei Sung

**Affiliations:** 1Department of Pathology, Changhua Christian Hospital, Changhua 50006, Taiwan; 28935@cch.org.tw; 2Department of Medical Technology, Jen-Teh Junior College of Medicine, Nursing and Management, Miaoli 35664, Taiwan; 3Department of Pathology, Chung Shan Medical University, Taichung 40201, Taiwan; jasmine.lyl@gmail.com; 4Department of Pathology, Chung Shan Medical University Hospital, Taichung 40201, Taiwan; 5School of Medicine, Chung Shan Medical University, Taichung 40201, Taiwan; william70325@gmail.com; 6Department of Medical Education, MacKay Memorial Hospital, Taipei 10491, Taiwan; 7Institute of Medicine, Chung Shan Medical University, Taichung 40201, Taiwan; 8Department of Urology, Chung Shan Medical University Hospital, Taichung 40201, Taiwan

**Keywords:** Krüppel-like factor 10, KLF10, prognosis, oral cancer, oral squamous cell carcinoma, overall survival

## Abstract

*Background and Objectives:* Krüppel-like transcription factor 10 (KLF10) plays a vital role in regulating cell proliferation, including the anti-proliferative process, activation of apoptosis, and differentiation control. KLF10 may also act as a protective factor against oral cancer. We studied the impact of KLF10 expression on the clinical outcomes of oral cancer patients to identify its role as a prognostic factor in oral cancer. *Materials and Methods:* KLF10 immunoreactivity was analyzed by immunohistochemical (IHC) stain analysis in 286 cancer specimens from primary oral cancer patients. The prognostic value of KLF10 on overall survival was determined by Kaplan–Meier analysis and the Cox proportional hazard model. *Results:* High KLF10 expression was significantly associated with male gender and betel quid chewing. The 5-year survival rate was greater for patients with high KLF10 expression than for those with low KLF10 expression (62.5% vs. 51.3%, respectively; *p* = 0.005), and multivariate analyses showed that high KLF10 expression was the only independent factor correlated with greater overall patient survival. The significant correlation between high KLF10 expression and a higher 5-year survival rate was observed in certain subgroups of clinical parameters, including female gender, non-smokers, cancer stage T1, and cancer stage N0. *Conclusions:* KLF10 expression, detected by IHC staining, could be an independent prognostic marker for oral cancer patients.

## 1. Introduction

Oral cancer is the sixth most common cancer in the world [1,2]. In spite of progress in research and therapy, the 5-year-survival rate has improved only minimally from 54% to 66% in the past 30 years [3,4]. Thus, prognostic tools that could promptly predict an unfavorable outcome of oral cancer are urgently needed for the early identification of potential outcomes and to modify existing treatment and follow-up strategies [5].

Carcinogenesis and tumor progression are believed to be substantially linked to the dysregulation of cell proliferation and the apoptosis resulting from cell mutation [6]. Krüppel-like-factor (KLF) family members are a group of transcriptional proteins containing three C_2_H_2_ zinc finger DNA-binding domains with a Krüppel linker structure between the zinc fingers. These proteins are involved in cell proliferation and the activation of apoptosis in normal tissues [7]. Krüppel-like transcription factor 10 (KLF10), also known as TIEG1, plays an important role in mediating the signaling of transforming growth factor beta (TGFβ), a multifunctional cytokine with a sophisticated mechanism involving the expression of cell cycle regulators, cell proliferation, and activation of apoptosis [8,9]. KLF10 participates in multiple biological processes and diseases, including the anti-proliferative process and differentiation control [8,9,10]. Moreover, elevated intracellular levels of KLF10 tend to mimic the apoptotic and anti-proliferative effects of TGFβ [11,12,13]. The detailed mechanism of the complicated signaling cascade has been widely investigated in several cancers for the purpose of using KLF10 as a reliable prognostic index of cancer progression [14,15,16,17,18,19], and the significant prognostic value of KLF10 as a biomarker for predicting the survival of patients with pancreatic adenocarcinoma has been demonstrated in previous studies [7].

The remarkable role of KLF10 in mediating carcinogenesis has aroused interest in predicting the clinical outcome of oral cancer. In the present study, KLF10 levels were measured by the immunohistochemical (IHC) stain analysis of oral cancer specimens. Whether KLF10 protein expression is associated with specific clinical features and survival outcomes was also investigated in this study.

## 2. Materials and Methods

### 2.1. Patients

In this retrospective study, our study investigated tumor samples from patients with primary oral squamous cell carcinoma (OSCC). The cancers were staged according to the Cancer Staging Manual of the American Joint Committee on Cancer. The clinicopathological features collected included gender, age, risk factors, histological type, and TNM stage (tumor (T), nodes (N), and metastases (M) satge) from the established database. The pathological diagnoses had been previously confirmed by two pathologists [20,21]. Those patients with missing clinical data or tissue loss during the IHC staining procedure were excluded from this study. The study was approved by the Institutional Review Board and Ethics Committee of the Changhua Christian Hospital, Changhua, Taiwan (IRB no. 131014) (3 March 2013). All methods were carried out in accordance with relevant guidelines and regulations.

### 2.2. IHC Staining and Scoring of KLF10

The IHC staining was performed at the Department of Surgical Pathology of Changhua Christian Hospital, Changhua, Taiwan, using the anti-human-KLF10 antibody (Santa Cruz, sc-23159; 1:75 dilution) as previously described [20,22]. The immunoreactivity scores were analyzed by the pathologists using scoring protocol as described previously [22,23], and the pathologists were blind to the clinical and prognostic data. A final consensus was obtained for each score by having all of the pathologists view the specimens simultaneously under a multi-headed microscope (Olympus BX51 10-headed microscope). The IHC staining scores were defined as cell staining intensity (0–3) multiplied by the percentage of stained cells (0–100%), leading to scores from 0 to 300 [20,22].

### 2.3. Statistical Analyses

The Student *t* test, and the χ^2^ test were applied for continuous or discrete data analysis. The associations between KLF10 expression and overall survival were estimated using the Kaplan–Meier method and univariate analysis and assessed further using the log-rank test [23,24]. Cox regression models of multivariate analysis were used to account for potential confounders with KLF10 expression fitted as an indicator variable. All the statistical analyses were conducted using SPSS statistical software (version 15.0; SPSS Inc., Chicago, IL, USA). All the statistical tests were two-sided, and values of *p* < 0.05 were considered statistically significant.

## 3. Results

### 3.1. Patient Characteristics

Overall, 286 patients (241, 84.3% male; 45, 15.7% female) with a mean age of 56 ± 11.5 years (range: 31–90 years) were enrolled in this retrospective study. The histological type of all the tumors collected was squamous cell carcinoma. Among the selected patients, several features were recorded and further categorized to observe the relationship between multiple factors and KLF10 expression. Overall, there were 117 (40.9%) smokers and 169 (59.1%) non-smokers. Fifty-eight (20.0%) patients were positive for betel quid chewing versus 228 (80.0%) patients who were negative. As for cancer stage, 56 (19.6%) patients were in the early stage (I) and 230 (80.4%) were in the late stages (II, III, IV). The overall 5-year survival rate was 58.5%, with a mean survival time of 4.7 years.

### 3.2. Correlation between KLF10 Expression and Clinicopathological Features

Representative images of the IHC staining of KLF10 are shown in Figure 1. The KLF10 expression score was 170 ± 65 (mean ± SD), and the median value was 90. Therefore, we defined a cytoplasmic KLF10 expression level <90 as a low expression. The relationships between KLF10 expression and multiple clinical parameters are listed in Table 1. Among all the parameters, male gender and betel quid chewing showed significant association with high KLF10 expression (*p* = 0.015 and *p* = 0.050, respectively), but no significant association was observed between KLF10 and other parameters, such as age, smoking, tumor differentiation, stage, T value, or N value.

### 3.3. Prognostic Value of KLF10 Expression in Oral Cancer Tumor Specimens

Univariate and multivariate analyses were used to evaluate the prognostic value of various parameters (Table 2 and Table 3). Overall survival constituted a major measurement in both analyses. The Kaplan–Meier survival curves demonstrated a relationship between patient prognosis and KLF10 expression (Figure 2). The univariate analysis indicated that an early stage (I) of oral cancer and a high expression of KLF10 were significantly associated with a better prognosis (*p* = 0.047 and *p* = 0.043, respectively). As anticipated, patients with early stage disease had a better 5-year survival rate than those at an advanced stage (II, III, IV) (72.2% vs. 55.2%, respectively; log rank *p* = 0.047). Moreover, a higher KLF10 expression was also linked with a higher 5-year survival rate as compared to a lower expression (62.5% vs. 51.3%, respectively; log rank *p* = 0.043). However, other parameters, such as age, gender, smoking, and betel quid chewing, had no statistically significant relationship with improved 5-year survival.

Multivariate analysis was performed to further determine whether KLF10 expression constituted an independent prognostic marker in our selected group (Table 3). After the adjustment of the confounding factors by a linear regression model, high KLF10 expression appeared to be the only parameter significantly correlated with longer mean survival. A high expression of KLF10 was associated with longer mean survival in comparison to low expression (4.9 vs. 4.2 years; hazard ratio (HR): 1.528, *p* = 0.035), but other parameters, including early stage, were not significantly associated with better mean survival. With further included T value and N value to the multivariate analysis, the prognostic role of KLF10 remained not changed (HR: 1.716, 95% CI: 1.154–2.551; *p* = 0.008).

### 3.4. Influence of KLF10 Expression on Overall Survival According to Clinical Parameters

A subgroup analysis was conducted using a multivariate method of determining survival outcome based on clinical parameters to more accurately identify the prognostic character of KLF10 expression (Table 4). Multivariate adjustment was performed for age, gender, smoking, betel quid, and cancer stage. Among these parameters, a significant relationship between high KLF10 expression and a higher 5-year survival rate was observed in a number of parameters, including female gender (88.6% vs. 44.9%; HR: 7.045, *p* = 0.016), non-smokers (62.2% vs. 49.9%; HR: 1.694, *p* = 0.04), cancer stage T1 (78.9% vs. 50.0%; HR: 3.074, *p* = 0.017), and cancer stage N0 (73.2% vs. 60.2%; HR:1.779, *p* = 0.043).

## 4. Discussion

In our study, we enrolled 286 patients with oral cancer and analyzed the histological expression of KLF10 in specimens removed from the patients. Multivariate analysis was performed to identify the relationship between multiple factors and the 5-year survival rate. The significant character of KLF10 in predicting the clinical outcome of oral cancer was discovered. Among the patients’ other documented data, a higher level of KLF10 expression under IHC staining was presented as the only factor correlating with a greater 5-year mean survival rate. In the subgroup analysis, we found that, in the groups of female patients, non-smokers, T1 stage, and N0 stage, a high KLF10 expression significantly correlated with a greater 5-year survival rate. In the parameter of T value, a high KLF10 expression was associated with a higher 5-year survival rate in the T1 subgroup (with a tumor smaller than 2 cm). On the other hand, the correlation was insignificant in the T2, 3, and 4 groups (with a tumor exceeding 2 cm). Therefore, it may be assumed that the prognostic role of KLF10 is more prominent in T1 lesions. Moreover, despite its statistical significance, the gender parameter should be interpreted cautiously because of the small sample (*n* = 45).

Several studies have broadly analyzed the probable mechanism and distinct importance of KLF10 in numerous types of cancer, such as pancreatic cancer, renal cell carcinoma, breast cancer, etc. [25,26,27]. However, there remained a lack of comprehensive data on the correlation between the degree of KLF10 expression and clinical prognosis, which prompted the present research to define the degree of correlation by quantifying KLF10 expression. As in previous studies, the prognostic role of biomarkers was assessed by using an immunoreactivity scoring system on pathological specimens to precisely describe the expression [22,28]. The present study, a novel model researching the relationship between high KLF10 expression and oral cancer prognosis, reproduced the scoring system to clearly address the clinical association. The majority of studies have aimed to identify the subtle character of KLF10 by discovering its sophisticated transcriptional pathway in modulating cancer progression [14,15,16,17,18,19], but few articles had investigated the direct connection between the degree of KLF10 expression and clinical outcome to elucidate the pivotal prognostic potential of KLF10 [7]. Hence, this study was designed to draw a practical clinical conclusion using 5-year survival as the primary outcome, which may aid clinicians in more precisely distinguishing a favorable prognosis from the opposite. Nevertheless, although the potential prognostic value of KLF10 was clearly demonstrated in our study, the detailed mechanism and molecular model of tumor suppression remain unclear and warrant further research.

The KLF transcription factor and another group of factors, known as transcription factor SP (specificity proteins), similarly contain three Krüppel-like zinc finger structures and are recognized as the SP/KLF family. This collective has been shown to participate in several cell functions, such as growth, apoptosis, differentiation, and angiogenesis. This illustrates that the SP/KLF family engages in multiple aspects of tumorigenesis [19]. Initially, *KLF10* was identified as an early gene induced by TGFβ and was named the TGFβ inducible early gene 1 (*TIEG1*) [29,30]. The TGFβ superfamily is a group of transcription factors that was discovered to have the function of mediating fundamental cell processes, such as proliferation, differentiation, death, cytoskeletal organization, adhesion, and migration. Its transcription of target genes is controlled mainly by SMADs (SMAD family members) proteins, a collection of intracellular mediators of the TGFβ family [31]. Serving as an effector protein of TGFβ-mediated cell growth control and differentiation, KLF10 is well known for its close relationship with TGFβ and consequent pivotal role in various cancers. KLF10 effectively represses cancer cell proliferation, with the overexpression of KLF10 reducing cell proliferation in many cancer types while its absence may enhance cell proliferation [19]. One previous study aimed to identify the clinical prognostic role of KLF10 in pancreatic cancer, and a higher expression of KLF10 was shown to be an independent predictor of progression-free survival and overall survival for pancreatic cancer patients [7].

Although the significant prognostic role of KLF10 was also identified in our study, several limitations warrant a cautious interpretation of the result. First, the pathological specimens were analyzed retrospectively after being resected from the tumor bed in a limited size, so it is possible that they were not sufficiently representative to demonstrate the protein expression of the whole tumor. Moreover, there were no data of adjacent normal tissue to compare paired tumor and normal expression. Additionally, the data were collected from patients of the same country, and the limited sample size may restrict the external validity. Moreover, a lack of information about cancer-specific death and adjuvant or neoadjuvant chemotherapy may also influence evaluation of the prognosis.

Much previous research has widely discussed the role of KLF10 in a transcriptional pathway that could potentially mediate tumorigenesis, cell proliferation, and apoptosis. Our study aimed more to directly identify the prognostic value of the biomarker and succeeded in demonstrating that a high KLF10 expression is associated with a more favorable clinical outcome in oral cancer. In light of the significant prognostic role of KLF10 shown in the present study and the few other studies that drew similar conclusions in relation to various cancers, subsequent studies are needed to develop new screening methods or therapies.

## 5. Conclusions

KLF10 expression could potentially be used as an independent prognostic marker in patients with oral cancer, especially those at the early T and N stages. However, due to the small sample size in our study, further research with larger populations is warranted to support our findings before its clinical application as a prognostic marker.

## Figures and Tables

**Figure 1 medicina-57-00017-f001:**
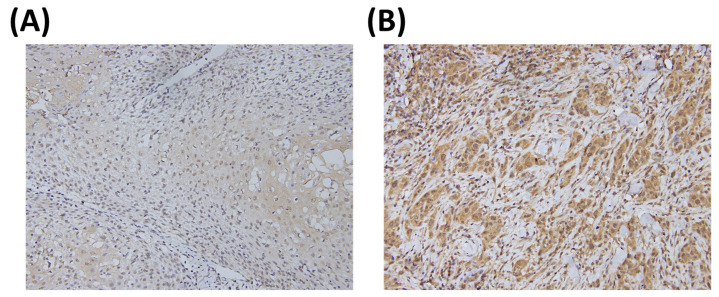
Representative immunostaining of KLF10 in oral squamous cell carcinoma (OSCC) specimens. The KLF10 expression levels were (**A**) low and (**B**) high.

**Figure 2 medicina-57-00017-f002:**
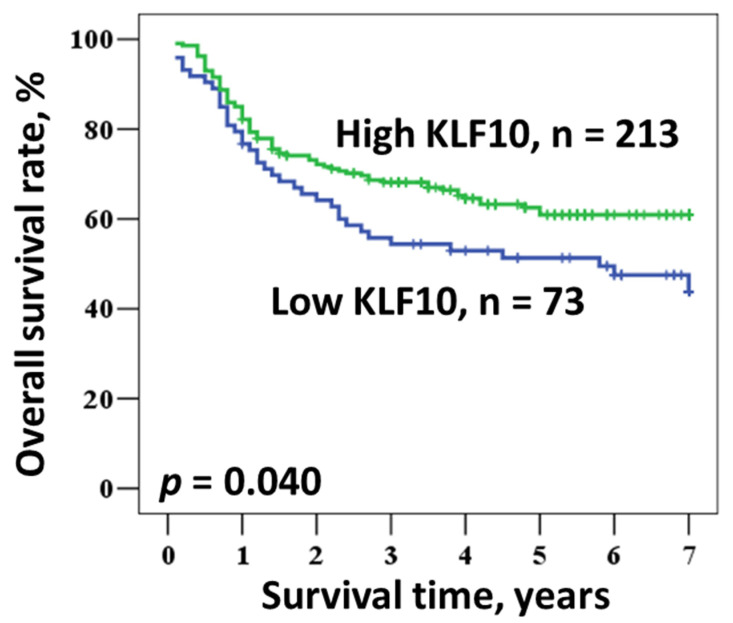
Kaplan–Meier survival of OSCC patients according to KLF10 expression.

**Table 1 medicina-57-00017-t001:** The relationships between Krüppel-like transcription factor 10 (KLF10) expression and clinical parameters in oral cancer patients.

		KLF10 Expression		
Parameters	Case Number	Low	High	*p* Value
Age (years)		55.7 ± 11.3	56.7 ± 11.5	0.519
Gender				
Female	45 (15.7)	18 (40.0)	27 (60.0)	0.015
Male	241 (84.3)	55 (22.8)	186 (77.2)	
Smoking				
No	169 (59.1)	47 (27.8)	122 (72.2)	0.287
Yes	117 (40.9)	26 (22.2)	91 (77.8)	
Betel quid chewing				
No	228 (79.7)	64 (28.1)	164 (71.9)	0.050
Yes	58 (20.3)	9 (15.5)	49 (84.5)	
Differentiation				
Good	45 (15.7)	11 (24.4)	34 (75.6)	0.856
Moderate + Poor	241 (84.3)	62 (25.7)	179 (74.3)	
Stage				
I	56 (19.6)	15 (26.8)	41 (73.2)	0.809
II + III + IV	230 (80.4)	58 (25.2)	172 (74.8)	
T value				
1	74 (25.9)	20 (27.0)	54 (73.0)	0.731
2 + 3 + 4	212 (74.1)	53 (25.0)	159 (75.0)	
N value				
0	177 (61.9)	48 (27.1)	129 (72.9)	0.431
1 + 2 + 3	109 (38.1)	25 (22.9)	84 (77.1)	

**Table 2 medicina-57-00017-t002:** Univariate analysis of the influence of various parameters on the overall survival of oral cancer patients.

		Overall Survival			
Parameter	Category	5-Year Survival (%)	Hazard Ratio	95% CI	*p* Value
Age	≥57/<57	58.8/58.5	0.980	0.679–1.415	0.914
Gender	Male/Female	56.8/69.0	1.447	0.812–2.577	0.210
Smoking	Yes/No	58.2/58.7	0.944	0.651–1.368	0.760
Betel quid chewing	Yes/No	60.1/57.9	0.846	0.527–1.358	0.489
Stage	II + III + IV/I	55.2/72.2	1.708	1.007–2.896	0.047
KLF10	Low/High	51.3/62.5	1.491	1.012–2.198	0.043

**Table 3 medicina-57-00017-t003:** Multivariate analysis of the influence of various parameters on the overall survival of oral cancer patients.

		Overall Survival			
Parameter	Category	Mean Survival (Years)	HR	95% CI	*p* Value
Age	≥57/<57	4.8/4.7	0.958	0.656–1.400	0.826
Gender	Male/Female	4.6/5.3	1.501	0.817–2.757	0.191
Smoking	Yes/No	4.8/4.7	0.884	0.562–1.389	0.592
Betel quid chewing	Yes/No	5.0/4.7	0.855	0.483–1.515	0.592
Stage	II + III + IV/I	4.6/5.5	1.698	0.988–2.918	0.055
KLF10	Low/High	4.2/4.9	1.528	1.031–2.265	0.035

**Table 4 medicina-57-00017-t004:** Multivariate analysis of the influence of KLF10 expression according to clinical parameters on overall survival in oral cancer patients.

	Overall Survival ^1^			
Parameter	5-Year Survival (%)	HR	95% CI	*p* Value
All cases	51.3/62.5	1.528	1.031–2.265	0.035
Age (years)				
<57	51.5/61.2	1.435	0.844–2.441	0.182
≥57	50.8/64.3	1.605	0.869–2.965	0.131
Gender				
Female	44.9/88.6	7.045	1.444–34.382	0.016
Male	52.7/57.8	1.289	0.833–1.997	0.255
Smoke				
Yes	53.6/59.0	1.262	0.651–2.447	0.490
No	49.9/62.2	1.694	1.024–2.800	0.040
Stage				
I	53.3/79.4	2.588	0.901–7.438	0.077
II + III + IV	50.8/56.6	1.395	0.903–2.157	0.134
T value				
1	50.0/78.9	3.074	1.219–7.755	0.017
2 + 3 + 4	52.0/54.8	1.260	0.801–1.982	0.317
N value				
0	60.2/73.2	1.779	1.018–3.108	0.043
1 + 2 + 3	34.1/41.6	1.725	0.970–3.065	0.063

^1^ Adjusted for age, gender, smoking, betel quid, and stage.

## Data Availability

The data presented in this study are available on request from the corresponding author. The data are not publicly available due to possible personal information breaches though they were de-linked.

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
