# Peer review of "High Expression of KLF10 Is Associated with Favorable Survival in Patients with Oral Squamous Cell Carcinoma"

_medicina, 2020, doi:10.3390/medicina57010017_

Round 1

Reviewer 1 Report

Could you include normal epithelium to compare it to cancer?

in your patient samples; why didn’t you compare between normal and abnormal epithelium?

overall it’s good manuscript but if you could add other methods to support IHC would be great.

 Thanks 

Author Response

Could you include normal epithelium to compare it to cancer?

in your patient samples; why didn’t you compare between normal and abnormal epithelium?

overall it’s good manuscript but if you could add other methods to support IHC would be great.

Answer:

Thank you for this suggestion. In current study, we have difficulty to include normal epithelium due to limited size of specimens. We used tissue microarray instead of whole-mount sections for this investigation and most of them did not contain normal tissue. We try to compare the expression of normal and tumor tissue via database which is funded by the Knut & Alice Wallenberg Foundation but the data was not available. We added this comment to our limitation. Thank you.

(line 228)

“Although the significant prognostic role of KLF10 was also identified in our study, several limitations warrant a cautious interpretation of the result. First, the pathological specimens were analyzed retrospectively after being resected from the tumor bed in a limited size, so it is possible that they were not sufficiently representative to demonstrate the protein expression of the whole tumor. Also, there was no data of adjacent normal tissue to compare paired tumor and normal expression.”

Reviewer 2 Report

Thank you for considering me for peer reviewing the paper “High Expression of KLF10 Predicts Increased Survival in Patients with Oral Squamous Cell Carcinoma” for the journal “Medicina”.

Briefly, in this prospective study KLF10 levels were measured to evaluate whether KLF10 protein expression is associated with specific clinical features and survival outcomes in oral cancer patients.

Please find below my comments:

INTRODUCTION

1) I would use this paper for oral cancer statistics; citation 1. Miranda-Filho A, Bray F. Global patterns and trends in cancers of the lip, tongue and mouth. Oral Oncol. 2020 Mar; 102:104551. doi: 10.1016/j.oraloncology.2019.104551.

2) “Having characteristics of delayed detection and consequent poor prognosis, it has long been a critical malignancy that warrants serious attention” line 2. Please reconsider use this sentence as it is not true completely.

3)The overall introduction is good, but the second paragraph needs to be reoriented. Specifically: “The recognized clinical features usually require intensive management, which severely affects quality of life and survival”. No clinical features above this sentence were explained. Please explain.

4) Despite in literature the role of KLFs are not argument of broad research, there is a very interesting and thorough article about KLFs (Physiol Rev 90: 1337–1381, 2010;doi:10.1152/physrev.00058.2009). In brief, by McConnell et al, the only KLF involved in the pathobiology of oral cancer is KLF-4 which is also associated to GI-tract malignancies. What is the rationale you used to evaluate the role of KLF-10 instead of KLF-4 which, does not seem to be involved in epithelial malignancies, but in pancreatic, bone, breast and blood cancers? Please thoroughly explain because it is a fundamental key in the article and important to the readership. Moreover, most of the studies you mentioned [6-18] do not report oral cancer and KLF-10 but mainly KLF-10 pancreatic adenocarcinoma, which is totally out of topic.  This needs to be changed accordingly.

MATERIALS AND METHODS

1) It is important for the readership to mention what kind of study you carried out (e.g. retrospective, prospective etc) at this point of the paper. 

2) Please mention WHERE this study was performed. Hospital? Dental School? Cancer center? Which department?

3) “Our study examined 286 tumor samples”. This is part of the results.

RESULTS

 1) Please add (%) at each numeric result. And viceversa.

 2) on table 1/results associated section it is not well explained the relationship between the expression of KLF-10 and gender (male/female) and betel quid (yes/no). In fact, on the table p-value has only reported once per Gender (overall, males+females?) and betel quid (yes/no); same for the other data. Specifically for gender for example, p-value 0.015 seems it belongs to “female gender” but as stated it belongs to males as reported statistically significant. Please correct properly and accordingly.

3) why HR and 95% Cl are not mentioned/expressed in table 1?

4) “A subgroup analysis was conducted using a multivariate method of determining survival outcome based on clinical parameters to more accurately identify the prognostic character of KLF10 expression (Table 4)”. Table 4 is not reported. Please adjust accordingly.

DISCUSSION

 1) “Oral cancer is the sixth most common malignancy globally, accounting for up to 2.1% of total cancer cases; 145,000 people suffered and died from cancer of the oral cavity and lip [24]”. This is a repetition of the introduction. Please change.

2) Overall the first section of the discussion should be shortened widely, as it is repetitive regarding the results of the study. A brief summary of the results should be included in the first section of the discussion, but it has to be no longer than 7-8 lines.

Author Response

Thank you for considering me for peer reviewing the paper “High Expression of KLF10 Predicts Increased Survival in Patients with Oral Squamous Cell Carcinoma” for the journal “Medicina”.

Briefly, in this prospective study KLF10 levels were measured to evaluate whether KLF10 protein expression is associated with specific clinical features and survival outcomes in oral cancer patients.

Please find below my comments:

INTRODUCTION

1) I would use this paper for oral cancer statistics; citation 1. Miranda-Filho A, Bray F. Global patterns and trends in cancers of the lip, tongue and mouth. Oral Oncol. 2020 Mar; 102:104551. doi: 10.1016/j.oraloncology.2019.104551.

Answer:

Thank you for this suggestion. We added the reference accordingly.

(line 40)

“Oral cancer is the sixth most common cancer in the world [1,2]. In spite of progress in research and therapy, the 5-year-survival rate has improved only minimally from 54% to 66% in the past 30 years [3,4]. Thus, prognostic tools that could promptly predict an unfavorable outcome of oral cancer are urgently needed for the early identification of potential outcomes and to modify existing treatment and follow-up strategies [5].”

2) “Having characteristics of delayed detection and consequent poor prognosis, it has long been a critical malignancy that warrants serious attention” line 2. Please reconsider use this sentence as it is not true completely.

Answer:

Thank you for this suggestion. We deleted that sentence.

(line 40)

“Oral cancer is the sixth most common cancer in the world [1,2]. In spite of progress in research and therapy, the 5-year-survival rate has improved only minimally from 54% to 66% in the past 30 years [3,4]. Thus, prognostic tools that could promptly predict an unfavorable outcome of oral cancer are urgently needed for the early identification of potential outcomes and to modify existing treatment and follow-up strategies [5].”

3)The overall introduction is good, but the second paragraph needs to be reoriented. Specifically: “The recognized clinical features usually require intensive management, which severely affects quality of life and survival”. No clinical features above this sentence were explained. Please explain.

Answer:

Thank you for this suggestion. We deleted that sentence.

(line 40)

“Oral cancer is the sixth most common cancer in the world [1,2]. In spite of progress in research and therapy, the 5-year-survival rate has improved only minimally from 54% to 66% in the past 30 years [3,4]. Thus, prognostic tools that could promptly predict an unfavorable outcome of oral cancer are urgently needed for the early identification of potential outcomes and to modify existing treatment and follow-up strategies [5].”

4) Despite in literature the role of KLFs are not argument of broad research, there is a very interesting and thorough article about KLFs (Physiol Rev 90: 1337–1381, 2010;doi:10.1152/physrev.00058.2009). In brief, by McConnell et al, the only KLF involved in the pathobiology of oral cancer is KLF-4 which is also associated to GI-tract malignancies. What is the rationale you used to evaluate the role of KLF-10 instead of KLF-4 which, does not seem to be involved in epithelial malignancies, but in pancreatic, bone, breast and blood cancers? Please thoroughly explain because it is a fundamental key in the article and important to the readership. Moreover, most of the studies you mentioned [6-18] do not report oral cancer and KLF-10 but mainly KLF-10 pancreatic adenocarcinoma, which is totally out of topic.  This needs to be changed accordingly.

Answer:

Thank you for this important information. Indeed, there was no article related to the prognostic role of KLF10 in oral cancer and that is the season our finding is worth for publication. Though KLF4 was selected to be the key KLFs in oral cancer in the reference of by McConnell et al., the prognostic role of KLF4 was not supported via TCGA database (as shown below, p=0.275). Interestingly, in the same database, the prognostic role of KLF10 in head and neck cancer is statistically significant (p=0.0289). The reason we focus on the prognostic role of KLF10 is due to elevated intracellular levels of KLF10 tend to mimic the apoptotic and anti-proliferative effects of TGFβ and TGFβ is already known to be associated with clinical outcome in oral cancer. For the last question, we know that the reference 6-18 are not related to oral cancer but there was no suitable reference in correlate KLF10 and survival of oral cancer patients. Thus, in this study, we provide our finding of this novel topic.

Considering there would be misleading if we add KLF4 in our manuscript and explain the reason why it was not chosen, we would like to keep our introduction and mainly focus on the KLF10. Sincerely apologize for no change of this nice comment.

Figure. The KFL4 expression is not associated with survival in head and neck cancer (data from TCGA database)

MATERIALS AND METHODS

1) It is important for the readership to mention what kind of study you carried out (e.g. retrospective, prospective etc) at this point of the paper.

Answer:

Thank you for this suggestion. We revised the section accordingly.

(line 70)

“In this retrospective study, our study investigated tumor samples from patients with primary oral squamous cell carcinoma (OSCC). The cancers were staged according to the Cancer Staging Manual of the American Joint Committee on Cancer. The clinicopathological features collected included gender, age, risk factors, differentiation, histological type, and TNM stage from the established database. The pathological diagnoses had been previously confirmed by two pathologists [20,21]. Those patients with missing clinical data or tissue loss while IHC staining procedure were excluded from this study. The study was approved by the Institutional Review Board and Ethics Committee of the Changhua Christian Hospital, Changhua, Taiwan (IRB No. 131014). ”

2) Please mention WHERE this study was performed. Hospital? Dental School? Cancer center? Which department?

Answer:

Thank you for this suggestion. The study was done at the Department of Surgical Pathology of Changhua Christian Hospital, Changhua, Taiwan, which obtained the certification of the College of American Pathologists. We revised the section accordingly.

(line?)

3) “Our study examined 286 tumor samples”. This is part of the results.

RESULTS

Answer:

Thank you for this suggestion. We revised the section accordingly.

(line 81)

“The IHC staining was performed at the Department of Surgical Pathology of Changhua Christian Hospital, Changhua, Taiwan, using the anti–human-KLF10 antibody (Santa Cruz, sc-23159; 1:75 dilution) as previously described [20,22]. The immunoreactivity scores were analyzed by the pathologists using scoring protocol as described previously [22,23], and the pathologists were blind to the clinical and prognostic data.”

1) Please add (%) at each numeric result. And viceversa.

Answer:

Thank you for this suggestion. We revised the section accordingly.

(line 107)

“Overall, 286 patients (241, 84.3% male; 45, 15.7% female) with a mean age of 56±11.5 years (range: 31–90 years) were enrolled in this retrospective study. The histological type of all the tumors collected was squamous cell carcinoma. Among the selected patients, several features were recorded and further categorized to observe the relationship between multiple factors and KLF10 expression. Overall, there were 117 (40.9%) smokers and 169 (59.1%) non-smokers. Fifty-eight (20.0%) patients were positive for betel quid chewing versus 228 (80.0%) patients who were negative. As to cancer stage, 56 (19.6%) patients were in the early stage (I) and 230 (80.4%) were in the late stages (II, III, IV). The overall 5-year survival rate was 58.5%, with a mean survival time of 4.7 years.”

(line 126)

Table 1. The relationships between KLF10 expression and clinical parameters in oral cancer patients.

KLF10 expression

Parameters

Case number

Low

High

p value

Age (years)

55.7±11.3

56.7±11.5

0.519

Gender

 Female

45 (15.7)

18 (40.0)

27 (60.0)

0.015

 Male

241 (84.3)

55 (22.8)

186 (77.2)

Smoking

 No

169 (59.1)

47 (27.8)

122 (72.2)

0.287

 Yes

117 (40.9)

26 (22.2)

91 (77.8)

Betel quid chewing

 No

228 (79.7)

64 (28.1)

164 (71.9)

0.050

 Yes

58 (20.3)

9 (15.5)

49 (84.5)

Differentiation

 Good

45 (15.7)

11 (24.4)

34 (75.6)

0.856

 Moderate + Poor

241 (84.3)

62 (25.7)

179 (74.3)

Stage

I

56 (19.6)

15 (26.8)

41 (73.2)

0.809

II+III+IV

230 (80.4)

58 (25.2)

172 (74.8)

T value

1

74 (25.9)

20 (27.0)

54 (73.0)

0.731

2+3+4

212 (74.1)

53 (25.0)

159 (75.0)

N value

 0

177 (61.9)

48 (27.1)

129 (72.9)

0.431

 1+2+3

109 (38.1)

25 (22.9)

84 (77.1)

2) on table 1/results associated section it is not well explained the relationship between the expression of KLF-10 and gender (male/female) and betel quid (yes/no). In fact, on the table p-value has only reported once per Gender (overall, males+females?) and betel quid (yes/no); same for the other data. Specifically for gender for example, p-value 0.015 seems it belongs to “female gender” but as stated it belongs to males as reported statistically significant. Please correct properly and accordingly.

Answer:

Thank you for this question. Table 1 is common table style for contingency analysis via chi-square analysis. In chi-square analysis, only a p value for a table with two rows and two columns (https://scholar.harvard.edu/files/lecture_19.pdf). Therefore, the p=0.015 was for both gender and both KLF10 expression. Thank you for understanding.

3) why HR and 95% Cl are not mentioned/expressed in table 1?

Answer:

Thank you for this question. As reference mentioned above (https://scholar.harvard.edu/files/lecture_19.pdf), there was no HR and 95% for table 1 which was analyzed via chi-square analysis. Thank you for understanding.

4) “A subgroup analysis was conducted using a multivariate method of determining survival outcome based on clinical parameters to more accurately identify the prognostic character of KLF10 expression (Table 4)”. Table 4 is not reported. Please adjust accordingly.

Answer:

Thank you for this comment. We revised the article accordingly.

(line 163)

Table 4. Multivariate analysis of the influence of KLF10 expression according to clinical parameters on overall survival in oral cancer patients.

Overall survival1

Parameter

5-year survival (%)

HR

95% CI

p value

All cases

51.3/62.5

1.528

1.031–2.265

0.035

Age (years)

 <57

51.5/61.2

1.435

0.844–2.441

0.182

 ≥57

50.8/64.3

1.605

0.869–2.965

0.131

Gender

 Female

44.9/88.6

7.045

1.444–34.382

0.016

 Male

52.7/57.8

1.289

0.833–1.997

0.255

Smoke

Yes

53.6/59.0

1.262

0.651–2.447

0.490

No

49.9/62.2

1.694

1.024–2.800

0.040

Stage

I

53.3/79.4

2.588

0.901–7.438

0.077

II+III+IV

50.8/56.6

1.395

0.903–2.157

0.134

T value

1

50.0/78.9

3.074

1.219–7.755

0.017

2+3+4

52.0/54.8

1.260

0.801–1.982

0.317

N value

0

60.2/73.2

1.779

1.018–3.108

0.043

1+2+3

34.1/41.6

1.725

0.970–3.065

0.063

1Adjusted for age, gender, smoking, betel quid, and stage.

DISCUSSION

1) “Oral cancer is the sixth most common malignancy globally, accounting for up to 2.1% of total cancer cases; 145,000 people suffered and died from cancer of the oral cavity and lip [24]”. This is a repetition of the introduction. Please change.

Answer:

Thank you for this comment. This sentence was deleted.

2) Overall the first section of the discussion should be shortened widely, as it is repetitive regarding the results of the study. A brief summary of the results should be included in the first section of the discussion, but it has to be no longer than 7-8 lines.

Answer:

Thank you for this comment. Indeed, the first paragraph is repetitive. We deleted the paragraph accordingly.

Reviewer 3 Report

Thank you for the opportunity to review this manuscript describing an Analysis on the prognostic significance of KLF10 in Primary Tumor tissue of oral squamous cell carcinoma specimens. The authors report on a large cohort of patients (n=286) and conducted a Survival Analysis including KLF10 Expression and several other Features.

The study is written appealingly and the structure is adequate. I think that a table is Need providing all Basic characteristics of the investigated cohort (including all relevant clinical and pathological data)

As it comes to the correlation analysis, the authors report that KLF10 Expression did not correlate with clinicopathological Features other than gender and betel quid chewing Habit.

Furthermore, in the Survival Analysis, only UICC (which is Sound) and KLF10 Expression were significant prognosticators.

This however seems odd, as in a proper Survival Analysis the authors should include T and N stage (most relevant prognostic factor in most available publications) and then include those factors for a multivariate Analysis. The fact that KLF10 was not correlated (negatively) with metastasis or T stage makes its prognostic role hard to believe or to interprete.

Author Response

Thank you for the opportunity to review this manuscript describing an Analysis on the prognostic significance of KLF10 in Primary Tumor tissue of oral squamous cell carcinoma specimens. The authors report on a large cohort of patients (n=286) and conducted a Survival Analysis including KLF10 Expression and several other Features.

The study is written appealingly and the structure is adequate. I think that a table is Need providing all Basic characteristics of the investigated cohort (including all relevant clinical and pathological data)

As it comes to the correlation analysis, the authors report that KLF10 Expression did not correlate with clinicopathological Features other than gender and betel quid chewing Habit.

Answer:

Thank you very much for the review of our manuscript. All the data were summarized in the Table 1 and we added the percentage in this revision. Thank you.

Furthermore, in the Survival Analysis, only UICC (which is Sound) and KLF10 Expression were significant prognosticators.

This however seems odd, as in a proper Survival Analysis the authors should include T and N stage (most relevant prognostic factor in most available publications) and then include those factors for a multivariate Analysis. The fact that KLF10 was not correlated (negatively) with metastasis or T stage makes its prognostic role hard to believe or to interprete.

Answer:

Thank you for this question. In current analysis, stage which was scored with TNM stage was adjusted. To answer your question, we reanalysis our data and the results remain not changed. However, the stage ends up to be not significant. We added this finding to the result section. Thank you.

Table. Multivariate analysis of the influence of various parameters on the overall survival of oral cancer patients.

Overall survival

Parameter

Category

Mean survival (yrs)

HR

95% CI

p value

Age

≥57/<57

4.8/4.7

1.115

0.762-1.632

0.574

Gender

Male/Female

4.6/5.3

1.888

1.026-3.475

0.041

Smoking

Yes/No

4.8/4.7

0.953

0.607-1.498

0.835

Betel quid chewing

Yes/No

5.0/4.7

0.922

0.523-1.625

0.779

Stage

II+III+IV/I

4.6/5.5

0.395

0.139-1.124

0.082

T value

2+3+4/1

4.5/5.4

2.407

1.034-5.605

0.042

N value

1+2+3/0

3.6/5.4

3.213

2.093-4.931

<0.001

KLF10

Low/High

4.2/4.9

1.716

1.154-2.551

0.008

(line 143)

“A high expression of KLF10 was associated with longer mean survival in comparison to low expression (4.9 years vs. 4.2 years; HR: 1.528, p=0.035), but other parameters, including early stage, were not significantly associated with better mean survival. With further included T value and N value to the multivariate analysis, the prognostic role of KLF10 remained not changed (HR: 1.716, 95% CI: 1.154-2.551; p=0.008).”

Round 2

Reviewer 2 Report

no more comments

Reviewer 3 Report

Dear authors,

thank you for the opportunity to review the revised Version of this manuscript on the prognostic significance KLF10 in oral Cancer. I feel that relevant improvements have been made to the style and the Content of the manuscript including the statistics. While originality is limited to the role of another retrospective biomarker study, the data has been drawn from a large cohort of patients and, thus, the results should be published.